# Managing a Curriculum Innovation Process

**DOI:** 10.3390/pharmacy8030153

**Published:** 2020-08-24

**Authors:** Jacinthe Lemay, Pierre Moreau

**Affiliations:** Department of Pharmacology and Therapeutics, Faculty of Pharmacy, Kuwait University, Jabriya 90805, Kuwait; andree.lemay@ku.edu.kw

**Keywords:** curriculum reform, change management, academic innovation, strategy formation, leadership

## Abstract

Curriculum reform is a long-term project that needs to be managed through detailed strategy. To create this strategy, the management team has to analyse the current situation by doing a thorough environmental scan and by identifying the gap between the current state and the desired program. To be implemented, the vision of the new program needs to rely on the generation of several potential avenues to come up with optimal solutions, likely involving some form of innovation. Indeed, to come up with the most promising ideas, there needs to be an environment conducive to reflection and experimentation. Throughout the phases of analysis, decision making and implementation, specific activities need to be organised to engage college members. Furthermore, such a profoundly impacting project needs to include a parallel change management strategy to account for expected human resistance, both individual and collective (internal culture). This article proposes a method and several concrete actions to help leaders and managers support the development and implementation of a new and innovative curriculum to promote and support advancement of local professional practice.

## 1. Introduction

Embarking on curriculum revision or reform is a major long-term project. When it also involves innovation in terms of the general structure of the program or the way in which education is delivered and assessment is conducted, it becomes an even greater challenge [1]. The success of reform will be influenced by several factors, including the current state of the curriculum, the local culture and support from higher management [2]. The main reason why curricular reform is challenging is that it involves all personnel working within a college. Moreover, the further the project moves away from known and predictable habits and customs (culture), the more it will bring resistance due to fear, uncertainty, perceived lack of competence and time requirements, amongst others [1]. Curriculum reform is a human challenge in addition to a technical one and needs to be managed as such. It is a matter of the heart and mind [1]. Failure to address the human aspect and local culture can create a situation where the project cannot be completed as intended. Indeed, there is a saying that “culture eats strategy for breakfast”—Peter Drucker.

Because curricular change is such a long-term project, measured by the time it takes to create the program plus the time to graduate the first cohort, it should prepare pharmacists for current but mainly future challenges, based on insight, experience and a good understanding of where the profession is going. In addition, it should incorporate as much pedagogical innovation as possible to make it relevant to professional needs and to last for a significant number of years beyond its launch date. Going through this lengthy process would suggest more radical changes, a real transformation, rather than tinkering with knowledge and skills by an incremental change [3].

Curricular reform is a significant endeavour, and its realization needs a structured long-term direction; it needs a strategy. As suggested by Bolman and Deal, “A vision without a strategy remains an illusion.” The strategy is the separation between thinking and doing [4], it is about choosing a path that is coherent with the vision and with the prevailing culture [5]. Forming a strategy is a process that requires 3 major phases [6]. First, one needs to scan the environment, to analyse and diagnose the current situation and the current strategic position, and to identify the drivers for change to develop the vision. Second, one must think of different solutions that have to be compared and contrasted to make the strategic and innovative choices supporting the vision, allowing the gap between the current and the desired state to be bridged. Third, the solutions need to be implemented so that the strategy is converted into action and the curriculum evolves towards the new vision.

According to the well-known Kübler–Ross curve [7], members of a college will go through a series of emotions during the project, as their level of competence is challenged and they are taken outside their comfort zone (Figure 1). From the current situation, they will start by being in shock, then by expressing denial and frustration and by being depressed. Finally they will experiment, accept, engage in and commit to the new situation. On a long-term project such as curricular reform, there will be a lot of time spent managing negative emotions until the general feeling returns to a more positive attitude. Constant and consistent communication and reassurance are key during the process. Thus, in parallel to the strategy related to curriculum reform itself, a second strategy needs to be focussed on managing change. This is necessary to give a chance to the project to succeed. Indeed, resistance to change is a normal phenomenon in any project that brings about any form of innovation [8]. Resistant individuals can use any weakness to derail such a demanding project. This means that the curriculum reform strategy has to be convincing, to be based on needs and to propose evidence-based solutions so that it is as robust as possible and cannot be easily dismissed. In addition, there needs to be a genuine concern and specific activities to manage change, tailored to the local context and the prevailing societal culture, as some societies are more eager to experiment and others are more conservative [9]. There are a number of change management frameworks that can help to prepare for this reality [8,10]. Throughout this article, change management activities that will support the three main steps of forming and implementing the main strategy will be presented. They are mainly based on the 8-step model of change management proposed by Kotter (Figure 2) [11].

Embedded in the curriculum reform and change management strategies are a series of curriculum development steps that one can follow to obtain the desired program. This can take many shapes and forms, and it is not the main focus of this article. To illustrate management of a curriculum reform in a way that is not too abstract, we will use the curricular development steps that we have previously reported [12]. Thus, this article presents the management of an innovative curriculum reform from the higher perspective of strategy formation that includes an embedded change management strategy allowing curriculum development and implementation according to logical steps of curriculum development (Figure 2).

## 2. Analysis of Drivers and Environmental Scan

### 2.1. Establishing a Sense of Urgency

Curriculum reform is based, first and foremost, on the recognition that there is a need to change and evolve the curriculum. The need to change may be triggered by various factors, such as new accreditation standards, change in the legislation, advancing pharmacy practice, competing institutions, evolving patient and population needs, etc. It is important for leaders, whatever the position in the organization, to keep abreast of these environmental changes and to communicate them to the members of the college. The intent is to create a sense of urgency for curriculum reform and to identify, remove or mitigate any source of complacency. Indeed, in the absence of an obvious major crisis that triggers reform, there is often complacency and it becomes critical to support colleagues in recognizing the need to change the curriculum and the importance of proceeding rapidly to avoid becoming obsolete [8]. There are various approaches to increasing the urgency level, for example, comparing against a gold standard or competing institution, presenting feedback from external stakeholders (e.g., students, healthcare professionals, government officials and patients) regarding their thoughts on the preparedness of the graduates, showing the opportunities and rewards that a curriculum reform would bring and the consequences of the status quo [11]. Another practical way of going about this is scheduling one-on-one meetings with college members to assess their own vision of the current situation and starting to educate them about the driving forces for the reform [8]. These meetings will enable leaders to develop a working relationship based on honest communication and trust, which may motivate the members to follow the lead and to collaborate on the reform. Indeed, one important aspect of people management and harnessing creativity is to know your own people [13].

### 2.2. Creating a Guiding Coalition

Such one-on-one meetings help identify individuals that are key supporters of the curriculum reform and that could make up your “guiding coalition” [11] because significant changes are challenging and require a team of motivated and committed people to initiate and implement the reform. An effective guiding coalition is based on three pillars: its members, trust and shared goals. When building a coalition, Kotter suggests keeping in mind four important characteristics to have an effective team [11]. The team should

have sufficient people in a position of power so that the progress made cannot be blocked by others;include individuals having various and complementary expertise to support optimal informed decisions. Therefore, including individuals from each department within the college is vital;consist of individuals with good reputations to give the team credibility and to ensure that college members accept the decisions made by the team; andinclude individuals with proven or recognized leadership skills to be able to drive the reform.

Here, an important distinction must be made between leadership skills, which provide vision and stimulates others to join, and managerial skills, which are needed to organize, plan and coordinate tasks [14]. The members must possess both types of skills and must work together within their own strengths for the benefit of the team.

An effective leadership style depends on the context and individuals in the team. A good leader should adapt to the situation and should adopt a style ranging from directing, coaching, supporting or delegating approaches as a function of individual’s developmental level [15]. With more junior or inexperienced staff, the leader needs to be more hands-on and to direct the work. In contrast, with college members experienced in curriculum reform, the leader can delegate and support the process [15]. 

In addition to having the right individuals, an effective guiding coalition is based on trust among its members. This may seem obvious but, often, teams are made of individuals from different departments who believe their loyalty must reside with their immediate unit and consequently may have a sense of rivalry towards others. Honest and transparent communication plays a key role in fostering trust, and activities aimed at stimulating open discussion may prove beneficial. For example, having activities that allow the team to pass the “forming and storming” phase (model of team dynamics [16]) by well-planned off-site events can be crucial for the team to start on the right foot. Further, having a vision and goals that unite the team is essential and each team member should deeply believe in and be committed to the shared vision and objectives for the team to achieve success.

### 2.3. Developing a Vision

Once a team is in place, the next important step is to collectively create a vision for the new curriculum which resonates with members of the college. This new vision should be based on information from a detailed scan of the environment and should start with a sound diagnosis of what needs to change in the curriculum and why. The diagnosis can include dimensions such as the external environment, the state of the educational “business” sector, the resources and capabilities at hand, the stakeholders, the governance and the culture of the organization [6]. Some useful tools to proceed with the environmental scan can be found in the Appendix A. External influences can be studied by using the PESTEL (political, economical, social, technological, ecological, legal) tool and stakeholder mapping, while internal factors can be analysed by the VRIO (value, rarity, inimitability, organizational support) tool and the cultural web. A SWOT (strengths, weaknesses, opportunities, threats) analysis addresses both external and internal factors but in a less detailed manner. Those tools are useful to analyse the current situation and to provide insights into potential external and internal enablers or roadblocks to support curriculum reform. A thorough environmental scan will help to understand the current state and to select the best solutions in the next step of the strategy formation. Organizational culture is critically important, as there is a saying that “culture eats strategy for breakfast”—Peter Drucker—which implies that, if one does not consider the internal culture of the organization, implementation of the strategy may fail. Further, examples of actions to help shift the culture to facilitate strategy implementation are discussed throughout the text and are summarized in Section 4.4.

Also as a part of the environmental scan, one needs to determine what stakeholders need and expect from pharmacists in order to develop a relevant needs-based education curriculum [17]. There are various ways to collect this vital information, namely focus group meetings and surveys. The stakeholder map already allowed identification of the major players involved in curriculum reform and the groups impacted by the role of pharmacists. Typically, the stakeholders include healthcare professionals (e.g., pharmacists, physicians, nurses, etc.), patients, students, healthcare administrators, and select political or university decision-makers. The objective is to organize meetings with a small number (8–10) of individuals per stakeholder group and to have a series of specific questions ready for discussion. This allows the college to capture the needs and concerns of every group of stakeholders. Such information can be complemented by a thorough survey of a larger group of stakeholders to collect more information about the professional services that are needed and expected from pharmacists. Collectively, these data will serve as the foundation to create and cement the vision for curriculum reform that will ultimately shape future pharmacists.

## 3. Making Decisions

### 3.1. Communicating the Change Vision

“Drugs don’t work in patients who don’t take them”—C. Everett Koop. Using a similar analogy, a vision is only a series of words on paper if it is not communicated, understood and endorsed by those involved [11]. It is important to achieve a sense of shared understanding and commitment towards the vision and its goals in order to generate motivation towards the transformation [1]. However, ensuring timely and clear communication is a tedious task. Effective communication of the vision should be based on simplicity and examples to illustrate its meaning. It should be repeated often using various forums (e.g., college or department meetings, monthly newsletter, etc.), and members of the guiding coalition should display behaviours that are aligned with the vision [11]. They should also be available to explain any perceived inconsistencies and to encourage a two-way communication of listening and understanding potential concerns that may arise among college members [1]. One can also create opportunities to communicate and have discussions about the vision such as town hall meetings, quarterly staff meetings or off-site retreats.

### 3.2. Empowering Individuals for Action (Part I)

The vision underlying the curriculum reform needs a strategy to come to life, and the strategy is often deliberate in nature, meaning that it arises from an intentional planning process coming from the college’s leadership team (or delegated to the guiding coalition) that is triggered or imposed by external factors such as accreditation standards, change in the legislation, advancing pharmacy practice and competing institutions. However, the selected strategy can be adjusted according to the college’s learning and experimentation, should foster innovation and should embrace new ideas as they come. In that sense, the initial strategy should be adaptable as it moves closer to implementation in order to make sure that all the good ideas are collected and eventually implemented [18]. 

#### 3.2.1. Leading Creativity and Innovation in Curriculum Reform

Once a sound vision is integrated and communicated, one can enter the strategy development phase. This relies on the premise that individuals are empowered to act with the goal of moving the agenda forward. One way to achieve this is to create subcommittees tasked with specific mandates (examples below). In the context of encouraging innovative curriculum reform, such committees should have three components needed for creativity: curriculum development expertise, task motivation and creative thinking skills [13]. Leading innovation and fostering creativity requires diversity either by tapping into the different disciplines within the college and/or by seeking external input. In addition to ensuring diversity and to have individuals participate in decision making, leaders must provide a safe environment to explore different creative options. Teams must have support from college leaders to introduce new concepts without risk of repercussions, especially in the early phase of brainstorming. Indeed, an innovation leader should have some creativity and should act as a facilitator by creating an environment that will favour innovation and let the “messy phase” of creativity take its course [19]. Once some solid concepts emerge, the leader must filter the options using a rational approach to make sure that the concept that is selected for further development is one that takes into consideration the environmental scan previously done, including stakeholders needs, social values and academic issues. Visionary and charismatic people are typically good at creative thinking, are self-reliant and inspiring, and may be best suited to leading innovation endeavours [20]. In addition to being creative, an innovation leader must be disciplined, process-oriented and able to determine which leadership style is needed to successfully develop an innovative curriculum, as previously alluded to [13].

#### 3.2.2. Selecting the Optimal Solutions

“The essence of strategy is choosing what not to do”—Michael Porter. Decision making starts with creating a constructive work environment that encourages individual contributions, motivation and commitment. Decisions can be challenging to make because often there are elements of uncertainty, complexity, and risk and human factors to consider. As such, using a systematic approach for decision-making may prove useful [21]. Horn et al. proposed the “innovation funnel” as a process-oriented model to drive innovation and to support optimal decision making [22]. Basically, the model involves a series of steps, such as idea generation, analysis, design, development and implementation, that are separated by screen checks before proceeding to the following step. This helps to define and clearly delineate the situation requiring a decision. When thinking about a problem or a situation, Liedtka proposes thinking from different perspectives or angles: (1) a high-level systems perspective; (2) a goal-oriented intentional focus; (3) the past, present and future to see a trajectory; (4) following the scientific method for being both creative and critical; and (5) embracing emerging ideas [23]. When used together, these five angles are complementary for making organisations more adaptable to change and they are useful in generating a number of valuable options to consider and analyse [23]. The SAFe tool described in the Appendix A may be helpful in helping determine the best option to select (Figure 3). SAFe stands for suitability, acceptability and feasibility evaluation. Before initiating the implementation process of the selected option, it is best to go back and to carefully review the information that leads to the decision. The intent is to ensure that the facts and information collected with the environmental scans and used to make the decision are reliable, thorough and accurate. If not, one may revert back to the analysis phase before making a final decision [18].

#### 3.2.3. Guiding Principles

The decisions taken have to be collected in a document that will serve as the master document to guide implementation of the curriculum reform and must be communicated in a clear, motivating and appealing manner. Decisions become statements or principles that need to be respected at all the steps of implementation by the different committees, working groups and course teams. We call this collection of decisions the guiding principles. It can start with the vision that was elaborated on earlier and can then expand according to domains (discussed below) for which decisions have been made before the curriculum can be constructed [12]. Failure to have a written guiding document, a sandbox delineating what is expected, can lead to a general drift from the initial intentions as the number of participants increases. Using the methods discussed above will help to come up with decisions that are “SAFe” to implement. 

The first domain is the high-level global curriculum characteristics that should be respected. This is where decisions regarding the curriculum type (knowledge-based vs. competency-based vs. hybrid), its general structure (linear vs. spiral education), the learning focus (active vs. passive learning), discipline integration (or juxtaposition), the general assessment approach, the relative importance of interprofessional education, practice laboratories, experiential learning and projects have to be made.

The second domain is concerned with the course development and the learning environment. It expands on the first domain and aims at providing more specific guidance for course teams to respect the essence of the vision and to produce the expected learning outcomes. It includes elements such as course content selection, integration of knowledge and practice, and the importance of problem-solving activities.

The third domain provides specific guidance towards the assessment which is an often-forgotten component of curricular reform. Since assessment drives learning [24,25], assessment is key for graduates to reach the expected outcomes. It should define the place of formative assessment, feedback, knowledge and competency (observation) assessment, and peer-assessment as well as the level of proficiency expected at different stages of the curriculum.

The fourth domain provides guidance for the use of technologies, so that there can be a more uniform virtual environment to support learning with appropriate technical support. It includes the potential use of an integrated learning management system, learning resources (books and periodicals) available for students, technological methods used in class or in laboratories (including electronic health record systems), online assessment tools, portfolios, and student requirements in terms of possessing a computer or tablet.

The fifth domain pertains to the rules and regulations that will guide admissions, transfers, progression and graduation. Each of the five domains, and others that one may find necessary, can be further defined in comprehensive frameworks, templates, prototypes and other tools to support the academic staff in their role and to offer a more homogeneous and predictable learning environment for students to focus on the content and not the “container” (see the next section). As discussed later, there needs to be coaching and education on methods or tools that innovate from the current state.

The guiding principles are the high-level decisions that dictate the overall intentions of the new curriculum. The next phase of decisions is to determine how this will translate into an actual curriculum. This highly depends on the previous choices made, and there are so many possibilities that it cannot be summarized here in terms of management. However, the guiding principles need to be respected or revised if their initial version proves to be incorrect. Basically, each principle has to be inserted within the curriculum. As an example, if the curriculum has been chosen to be competency-based, there should be moments within the curriculum allowing students to practice and be assessed on their performance of delivering expected professional activities. Also, because competence is based, in part, on being knowledgeable, the essential knowledge for each competence has to be identified and inserted within the curriculum. If guiding principles include disciplinary integration, the coursework has to reflect this by having theme-related courses that involve several disciplines. For more examples on the enactment of guiding principles into an actual curriculum, we have recently published our own experience [12].

## 4. Implementation

### 4.1. Empowering Individuals for Action (Part II)

Now that the curriculum is developed and communicated at college and university levels to obtain the necessary approvals, it is time to work on the last phase of the strategy: implementation. This phase is not trivial and requires strong support from the college administration to allow its members to have protected time to work on the new curriculum in parallel to their education, administrative and research tasks. There are several issues that can lead to suboptimal execution of implementation, including lack of leadership, poor communication and conflicting priorities within an ambiguous strategy [26].

To make sure that communication flows and that the staff feels supported throughout the implementation phase, an implementation support team (IST) should be created to provide constant guidance, to help implement the learning methods and to review the courses to make sure they align with the guiding principles (Figure 4). The academic staff needs to feel empowered to reach the expected goals by being trained and coached. Letting the implementation go unattended will most likely result in a curriculum that will resemble the current one and not the one envisioned. When people do not get the support needed to move beyond their comfort zone, most will naturally revert to what they are currently doing for fear of failure or being criticized, with the exception of a handful of more adventurous individuals. 

The IST can be an evolution and expansion of the guiding coalition and needs to represent all departments or scientific sectors of the college. It should consist of influential colleagues having a clear desire to make the transition a success. If new members are joining the initial team, they have to be brought up to speed to understand the choices that were made and the reasons why. It is not a time to reconsider major changes, although one can always see if suggestions from new team members make sense to improve the current strategy. 

It is also important to emphasize that not everything has to be fully planned before moving forward. Some things can be decided along the way after implementation has started, as they may depend how initial decisions actually turn out. Indeed, “If we wait until we are ready, we’ll be waiting for the rest of our lives”—Lemony Snicket. Another important concept to keep in mind is the “Pareto Principle”, also referred to as the “80–20 rule” [27]. This principle is based on the fact that roughly 80% of results come from 20% of the efforts. Thus, focusing on fewer key elements can provide most of the benefits. This principle can be applied in several contexts and can help redirect efforts when perfectionism becomes paralyzing. 

In addition to the IST, additional working groups should be created to support other aspects of the curriculum (Figure 4). This is to offload the main support team and to involve more players in implementation of the curriculum. When people are actively involved in a project, they usually offer less resistance to change [28]. For instance, if active learning is to be heavily used within the new curriculum, a specific working group can be created to support the academic staff by developing training workshops on the main active learning approaches. These workshops will go a long way in preparing and supporting staff to develop their learning material. It will also make sure that the student experience will be more homogenous than if the staff is left to implement the active learning methods on their own without proper guidance. A similar approach can be used for assessment (including objective structured clinical examination (OSCE) and competency assessment) and other general aspects of a curriculum, as identified in the guiding principles.

Experiential learning is becoming a more substantial component of modern pharmacy curricula [29]. It is an area where a dedicated team can work on the placements’ learning objectives, site recruitment (capacity building), preceptor training and placement allocation policies. This also expands the number of staff involved in curriculum implementation (Figure 4). They can report their work to the IST to keep the desired alignment with more theoretical and practical components of the curriculum. Ideally, such a team should include individuals representing all aspects of the placement management, from academic staff to support staff, including preceptor and student representatives.

Importantly, every working group should develop “SMART” objectives, meaning that the objectives are specific, measurable, attainable, relevant and time bound [30]. The benefits of SMART objectives include that they are unequivocal and clear and provide the foundation for key performance indicators (KPIs) and specific metrics which allow the team to measure, track and report progress. For example, a SMART objective for an experiential learning working group could be to recruit and train 5 new preceptors in 5 different healthcare institutions every 6 months. The associated KPIs could be the number of preceptors recruited as a percentage of the target 25. Furthermore, a simple system of green, yellow and red lights can be used to illustrate whether the team is on target, experiencing some delay or significantly behind.

Finally, it is also important to create a program committee that will oversee all administrative aspects of the implementation (Figure 4). The program committee can evolve from the IST or ideally have some members that are part of both to ensure optimal communication/sharing of information (avoiding things “falling through the cracks”). This committee can start to work quite early in the process to devise a working schedule for all years of the program at once as the older program is phased out. It can also work with institutional bodies for admissions and registration. They can start preparing a model of continuing program improvement, using standing entities like a quality assurance unit and a curriculum committee.

For each course within the new curriculum, there should be one team leader for content and assessment development, often called the course coordinator. Even if teams are expected to work together for a single course, it is generally a good idea to have a single person responsible for ascertaining accountability and leadership. Sometimes, when a team is composed of several co-coordinators, nobody actually feels accountable towards implementation or there is confusion regarding allocation of specific responsibilities. For integrated courses, one has to make sure to include all relevant disciplinary experts, but their involvement should be proportional to the amount of material that will be part of the course. As such, some team members may not attend all meetings if their presence is not necessary in an effort to reduce the workload of course development. For some transversal competencies, like communication, critical appraisal and problem solving, to name a few, it can be a good idea to identify a content expert that can be used as a consultant for several courses to ensure that competency is gradually being mastered by students throughout the academic years through activities that are gaining complexity. 

There should be a mechanism to resolve conflict or inertia within teams if the members are struggling to do it by themselves. Ineffective teams may miss deadlines and cause problems for implementation of the curriculum. This is another reason to have an IST that schedules regular meeting with course team leaders. Workshops on team dynamics and effective teamwork can be a good investment to help the teams move beyond the storming phase into the performing phase [16]. Moreover, for course teams that do not provide the expected results, the IST has to determine if it is related to resistance to change or other factors, and to address the issue quickly. Actions have to be taken early on to prevent major fallbacks, and it may involve changing some members of a course team to allow a more productive environment. The dean and vice-dean of academic affairs could be involved in this aspect to support the general implementation effort. 

### 4.2. Generating Quick Wins

The next component in Kotter’s model of change management that fits within the implementation phase of the strategy is to generate and celebrate short-term visible improvements or “quick wins” [11]. This creates a major boost in the morale of those who have gone through the innovation implementation, showcases progress and successful achievements, and will reassure those still awaiting their turn to contribute to the new curriculum. As much as possible, these quick wins should be planned ahead and not left to chance. For instance, a student satisfaction survey can be prepared in advance for the first cohort of students that have undergone active learning within their first semester. Other examples include positive feedback from preceptors, healthcare providers and administrators working closely with students from the new curriculum.

The quick wins can also be generated outside the new curriculum before its actual implementation, for example by applying some of the planned innovation within the existing curriculum and by reporting its satisfaction. If the college offers a shorter program with a small number of students, like an add-on PharmD or a residency program, it can also become a testing ground for some of the innovation choices. One has to keep in mind that the solution needs to be scalable for the main pharmacy undergraduate cohort. Moreover, students that are used to one type of teaching approach may not adapt readily to a new form of education and can report a negative experience because, in the short term, they cannot see the positive outcomes and rely mainly on their feelings [31]. 

Celebrating the quick wins can take many forms. It can be publishing a manuscript on the project and its innovation, supporting presentations at educational conferences, or sharing academic staff and student experiences within the college through internal communication channels. Students can also be encouraged to write in their association’s journal or on their social media platform to share their positive experience. 

At this step, it is important to communicate with stakeholders so that they can be reassured that the project is progressing well and that positive outcomes are emerging early on. Importantly, the nature, frequency and type of communication will depend on their position on the stakeholder map. Social media is a great way of providing regular and focused communications with the pharmaceutical ecosystem without needing to resort to long face-to-face meetings and presentations. The goal is to create a positive momentum around implementation of the new curriculum. Leaving a communication void may provide the necessary space for naysayers to distort the narrative. It might prove helpful to develop a thorough communication plan either with the support of the college’s public relations office or with an external communications consultant.

As previously mentioned, key performance indicators (KPI) should measure your “SMART” objectives to show the impact of the new curriculum beyond subjective experiences. Improvement in student grades, reduced attrition rates, global student satisfaction measurements and other indicators can convince the more analytical colleagues that the new approach is working. Impact on the quality of the admitted students is also another potential metric to measure, as positive experiences from students quickly circulate on social media and this can lead to better students wishing to enrol in an innovative and stimulating professional program.

### 4.3. Building on Change

Until the curriculum reform is completely implemented, the IST needs to supervise the work being done. Ideally, there should be a generous global timeline to allow each course team to have plenty of time to work on the syllabus and to define their learning and assessment strategies according to the guiding principles. Workshops can be done in phases, with colleagues newly involved having their training only just before they need it. Involving colleagues who have already gone through their course implementation in the subsequent workshops can help to alleviate any anxiety related to planned innovation. This approach keeps the momentum, allows to share experiences and to build on change.

It is also a good idea to organize a yearly town house meeting to summarize the progress and to discuss the plan ahead, as the curriculum is gradually implemented. On a more regular basis, the college can publish newsletters to share learnings, emphasizing what worked and what did not. A “lunch and learn” activity can also be created and supported by higher management, where new experiences are shared and discussed. Such a community of practice is an effective way of identifying and refining best practices for some aspects of learning and assessment identified in the guiding principles. These activities can be organized by the working groups already identified (Figure 4). If the institution has specific interest groups, academic staff should be motivated to attend and share the positive outcomes at this level. Recognition by the institution of the college as a leader and game changer will most likely be acknowledged, and the accolade will “rain down” on all its members. It is the same in the context of conferences and scientific societies, where recognition of the college’s leadership leads to improved notoriety and disciplinary ranking. Taken together, these will foster a sense of pride and will reinvigorate the motivation and engagement of college staff.

As the reformed curriculum is gradually implemented, changes may be required at the college level to sustain the new learning environment. It is indeed important to put the needed structure in place to ensure that the forming curriculum does not revert back to a mix of old and new because the necessary support is not present. For instance, the experiential learning working group can become an official unit with the task of overseeing this component of the curriculum, with the necessary technical and academic workforce (Figure 4). Also, if competence assessment is to be done, a competence review unit can be instituted to closely follow students’ performance and to provide remedial activities as early as possible. A quality assurance and accreditation unit can work towards continuing improvements and aligning the new curriculum with accreditation standards by identifying gaps or tools to be developed. If interprofessional education is introduced in the new curriculum, a global unit, including members from other health sciences, needs to be put in place to support this component of the curriculum. Basically, the new structure should support the major novel components of the curriculum, as outlined in the guiding principles. This may require targeted hiring of academic staff and personnel if it was not already done during the early implementation phase.

### 4.4. Imbedding Change Into the Culture

The organizational culture is a complex and living entity with technical and social customs [6] that are shared through values, attitudes, beliefs and assumptions [32]. It is an organization’s personality. Culture is important because it guides the ways in which employees think, feel and act; it guides decisions and performance [33]. It is based on the group’s past accumulation of shared learning, problems solved, adaptation to external situations and internal integrations, usually retaining what worked well and reproducing similar behaviours for new challenges [33]. As such, culture usually changes when experiences are positive for the whole organization. Success of implementing innovative solutions allows for a certain opening for future experimentation when facing new challenges. Indeed, an organization’s capacity to change is influenced by elements such as attitude towards criticism, mindset towards experimentation, willingness to give autonomy and support, and openness to discuss sensitive issues [34]. Culture change is a slow process, and one should be careful if trying to transform the culture too quickly [6,33]. A strategy that is not aligned with the prevailing culture will have a harder time being implemented by “going against the grain”. Moreover, a negative experience can make any future attempt even more difficult, as the collective memory of the previous attempt will quickly resurface and will enhance resistance. This is one of the reasons why curriculum reform needs an effective strategy and celebration of improvements, as discussed previously.

The academic culture in general is one that has been quite static in the past decades, and several authors recognize this status [1,35,36]. Universities are hierarchical organizations, and they tend to be internally focused and less inclined to innovation (besides its research mission) as compared to, say, technological industries, which are more adhocratic [37]. Academia is still very professor-focussed and has not embraced a shift towards student-centred learning despite the benefits supported by educational research [38]. It is similar to the healthcare system that is slowly moving from a physician or provider-focussed system towards patient-centred care. However, the shift seems even slower in academia, and it is lagging in providing the work-ready workforce needed [36]. This shows how a culture is strong and impacts attempts to move some of the fundamental beliefs. However, the social accountability of universities is coming under pressure and changes are happening to produce graduates that better address societal needs in a more efficient way [36,39]. It is important to recognize this aspect as one engages in reform of a professional curriculum. It is an integral part of the environment that a college has to consider, and it could even be the main driver for change, as discussed in Section 2.

To support innovation and a gradual shift in culture, leaders have to use certain mechanisms, most of which have been discussed already, like (1) being involved in the transformation by adopting the appropriate leadership style; (2) allocating resources to it; (3) coaching, rewarding and promoting people aligned with the transformation; (4) changing the organizational structure; (5) hiring new people; (6) changing physical allocations; and (7) developing new procedures to support innovation [33]. Importantly, identifying and supporting the future leaders of the college is important to sustain an open mindset towards innovation and change. Indeed, part of gradually changing the culture is to reward, coach, train and promote those that are aligned with the new direction, sending a message of what is valued. Academics are more or less prone to supporting innovation and having a succession plan that includes people that are not afraid of change is important to imbed innovation into the prevailing culture in a sustainable manner.

## 5. Conclusions

The success of curriculum reform hinges on a sound strategy that starts with thorough analysis of the current local environment, continues with generating options and making sound choices to bridge the gap appropriately, and finishes with a strong implementation plan. It is all about identifying the current and anticipated gaps and working to address them. For such a lengthy project, the strategy has to be flexible enough to incorporate up-to-date advances and innovation to ultimately reach the vision with a long-lasting effect. 

In order to best serve the community, graduates need to be relevant and trained to be work-ready, and in most cases, this involves innovating the way education is delivered and assessed. The college management team should create an environment that fosters innovation and leaders adopt a style that aligns with it. One has to remember that, by the time the first cohort graduates, several years will have passed since the curriculum was developed. Therefore, the design of the curriculum should be flexible, adaptable and “future-proof”. Curriculum reform is a major endeavour that cannot be repeated often. It is a global project involving the whole college, and it can become an amazing collective experience if it is planned and managed appropriately.

All along the way, change management has to be organized to make sure that the members of the college are gradually engaged in this global project. Managing change is managing people; leaders and managers need to have good emotional intelligence to identify the sources of change resistance and to address them diligently and effectively [40]. This is true at the individual and collective levels. Indeed, the internal culture is a powerful element that can derail great projects if it is not considered initially or inappropriately gauged. The positive side is that, if the culture is understood and respected, it can become a great ally to engage members of the college in an amazing journey that will make them proud of their new curriculum and their college’s reputation for years to come.

## Figures and Tables

**Figure 1 pharmacy-08-00153-f001:**
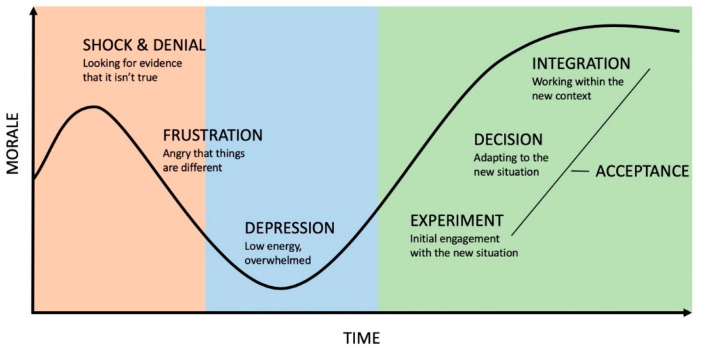
Kübler–Ross model of the emotional impact of change, based on their grief cycle [7]. Adapted from several sources.

**Figure 2 pharmacy-08-00153-f002:**
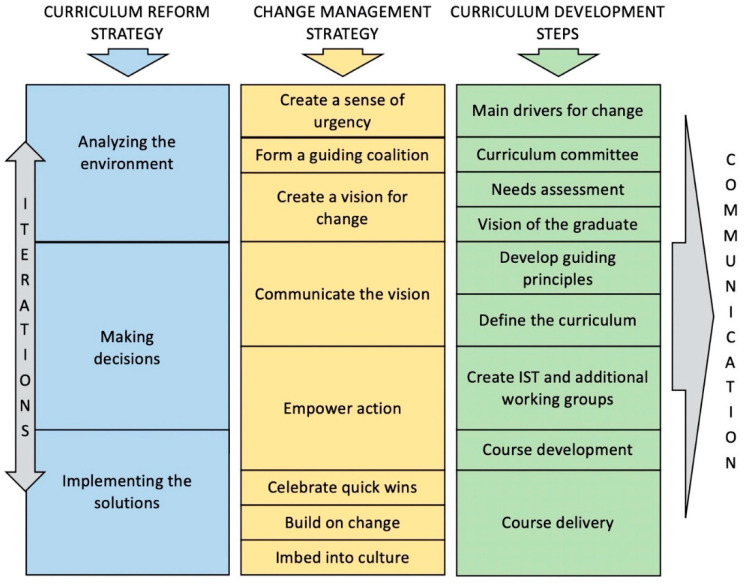
Parallel processes of (1) the curriculum reform strategy, (2) the change management strategy [11] and (3) the curriculum development steps. Curriculum development steps can vary, but those described are based on previous experience [12]. Although depicted as linear, from top to bottom, the processes allow for some iterations when downstream steps uncover flaws in upstream assumptions or decisions or when new information becomes available. Communication is important throughout the curriculum reform. IST: Implementation support team.

**Figure 3 pharmacy-08-00153-f003:**
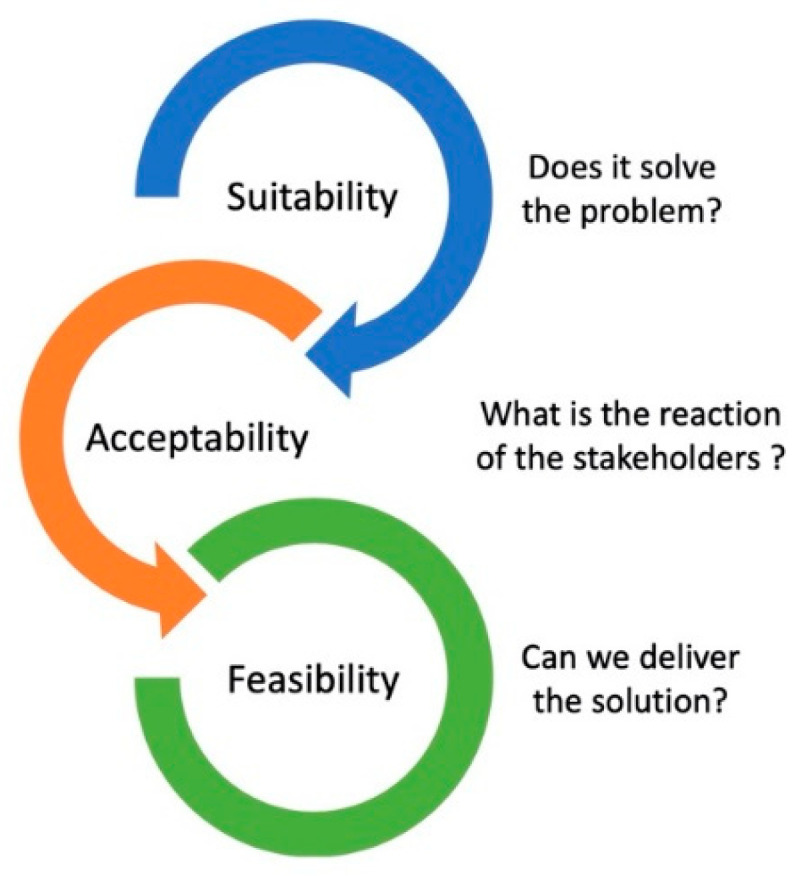
Evaluation of strategic options using the SAFe (suitability, acceptability and feasibility evaluation) framework [6].

**Figure 4 pharmacy-08-00153-f004:**
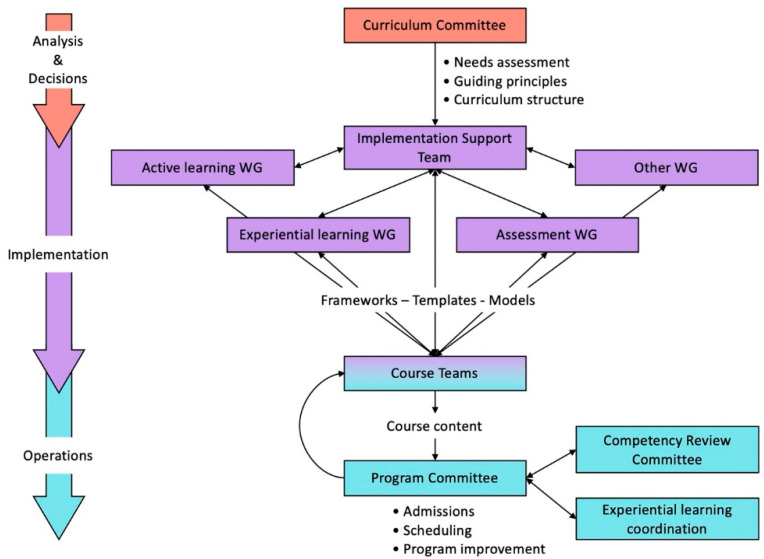
Example of committees and working groups (WG) that can assist in the strategy formation, its implementation and the daily operations: recruiting more individuals into the project improves engagement and limits resistance. Some deliverables (unboxed text) are mentioned as an illustration of the output of the different committees. Some implementation working groups can be later morphed into operational committees or communities of practice.

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
