# Peer review of "Managing a Curriculum Innovation Process"

_pharmacy, 2020, doi:10.3390/pharmacy8030153_

Round 1
Reviewer 1 Report
Very well written manuscript that is sure to help leaders, managers, and others in curriculum innovation.
Author Response
Thank you very much for your point of view on our manuscript
Reviewer 2 Report
Overall, this reviewer finds this manuscript redundant and wordy. The authors pull theory from multiple sources on change management and spend a considerable amount of time on how to manage faculty when embarking on a curricular revision in a pharmacy school. This is theory that is taught in master level management courses that address change and how to manage that change in academia. The reviewer did not find that this manuscript adds to the conversation.
If the editors disagree with this review, at the very least the authors need to reduce the text significantly. There is little insight here, therefore it should be written as a short review and the authors should also reference the relevant textbooks used to teach this information. In addiiton, the paragraphs need to be structured better with effective transitions.
Author Response
Thank you very much for taking the time to read our manuscript.
We regret that you did not find our work of value. However, there are not a lot of references out there that propose specific activities to help with the development and implementation of a curriculum in pharmacy (or in other disciplines). However, we did reference the major reference books on the topics that we addressed. Your comments were not specific and difficult to implement, as they could imply an unlimited number of possibilities. In that context, we kept the structure as initially proposed.
Reviewer 3 Report
Revising the curriculum is a daunting task for university members and a common problem in the world. In fact, their leaders have to make adjustments to every situation for these jobs, so they are considered to have heavy responsibility and stress.
This paper presents the importance of change management strategy when revising and reforming the curriculum. In addition, the authors is trying to use some tools (PESTEL, SWOT, CULTURAL WEB, etc.) and It helps readers understand how to control process management. Of particular interest to me was the author's focus on humans and culture.Such work is interesting to a lot of faculty member, it would be gives them meaningful information.
There is no doubt about the quality of the work presented, but I would appreciate if you reconsider the following points.
Line150 (2.3 Vision Creation)
How did you decide the order in which some tools (PESTEL, VRIO, SWOT, etc.) are listed? Does the current order have any special meaning?
Generally, people use VRIO after other external analysis such as PESTEL and SWOT. Therefore, we recommend changing the order as follows: 1) PESTEL, 2) SWOT, 3) VRIO.
Line225 (CULUTURAL WEB)
This is an advice to make it easier for readers to imagine how to use CULTURAL WEB.
You may want to add more about what to do if the strategy doesn't fit the culture of their organization.
Line341 (Selecting the best solution)
The title of this chapter is "Selecting the Best Solution". However, the paragraphs that follow indicate that you "create a document on guiding principles." Is there any reason to explain these two parts together? Unless you have a specific reason, it seems appropriate to separate up to line 340 and after line 341.
Author Response
Thank you for your constructive review of our manuscript. We addressed your more specific suggestions:
1- Line 150 (2.3 Developing a vision)
The order that we chose was to propose a tool to analyse the external environment thoroughly (PESTEL), then the internal capabilities (VRIO), before bringing the SWOT analysis which is a mix of both external and internal factors. SWOT is often criticized as being more superficial, so we preferred to introduce more detailed methods before introducing the SWOT analysis. We modified the text slightly to reflect this aspect.
2- Line 225 (Cultural Web)
The impact of culture on the strategy implementation is discussed later in the manuscript. Around line 254, we added a sentence to refer the reader to section 4.4 where practical examples of actions to be performed to shift culture are summarized. Also around line 589, a sentence was added to warn against the danger of “going against the grain”
3- Line 341 (selecting the best solution)
It is a very good idea to separate the decision making process from the document that contains these decisions. We have added an additional sub-section (subsection 3.2.3 Guiding Principles) to reflect that.
Round 2
Reviewer 2 Report
This manuscript contains only minor changes in grammar. The content continues to be less than novel and can be found in any curriculum change and management textbook. This reviewer does not believe that it contains any original ideas from the authors.
Author Response
We thank you for your comments. It is unfortunate that you have this point of view on our manuscript. It includes a lot of ideas and examples to bring the theory into practice. This is not information that is readily available. We could not find it anywhere in any case. In this second revision, we have reduced the text by 1028 words as initially requested and moved the tool description in supplemental format.